# Upregulation of p75NTR by Histone Deacetylase Inhibitors Sensitizes Human Neuroblastoma Cells to Targeted Immunotoxin-Induced Apoptosis

**DOI:** 10.3390/ijms23073849

**Published:** 2022-03-31

**Authors:** Simona Dedoni, Alessandra Olianas, Barbara Manconi, Maria Collu, Barbara Tuveri, Maria Elena Vincis, Maria C. Olianas, Pierluigi Onali

**Affiliations:** 1Department of Biomedical Sciences, Section of Neuroscience and Clinical Pharmacology, University of Cagliari, 09042 Monserrato, Italy; dedoni@unica.it (S.D.); mcollu@unica.it (M.C.); btuveri@unica.it (B.T.); vincis@amm.unica.it (M.E.V.); mariolina.olianas@gmail.com (M.C.O.); 2Department of Life and Environmental Sciences, Section of Biochemistry, University of Cagliari, 09042 Monserrato, Italy; olianas@unica.it (A.O.); bmanconi@unica.it (B.M.)

**Keywords:** entinostat, valproic acid, p75NTR, anti-p75NTR immunotoxin, apoptosis, human neuroblastoma cells

## Abstract

Histone deacetylase (HDAC) inhibitors are novel chemotherapy agents with potential utility in the treatment of neuroblastoma, the most frequent solid tumor of childhood. Previous studies have shown that the exposure of human neuroblastoma cells to some HDAC inhibitors enhanced the expression of the common neurotrophin receptor p75NTR. In the present study we investigated whether the upregulation of p75NTR could be exploited to render neuroblastoma cells susceptible to the cytotoxic action of an anti-p75NTR antibody conjugated to the toxin saporin-S6 (p75IgG-Sap). We found that two well-characterized HDAC inhibitors, valproic acid (VPA) and entinostat, were able to induce a strong expression of p75NTR in different human neuroblastoma cell lines but not in other cells, with entinostat, displaying a greater efficacy than VPA. Cell pretreatment with entinostat enhanced p75NTR internalization and intracellular saporin-S6 delivery following p75IgG-Sap exposure. The addition of p75IgG-Sap had no effect on vehicle-pretreated cells but potentiated the apoptotic cell death that was induced by entinostat. In three-dimensional neuroblastoma cell cultures, the subsequent treatment with p75IgG-Sap enhanced the inhibition of spheroid growth and the impairment of cell viability that was produced by entinostat. In athymic mice bearing neuroblastoma xenografts, chronic treatment with entinostat increased the expression of p75NTR in tumors but not in liver, kidney, heart, and cerebellum. The administration of p75IgG-Sap induced apoptosis only in tumors of mice that were pretreated with entinostat. These findings define a novel experimental strategy to selectively eliminate neuroblastoma cells based on the sequential treatment with entinostat and a toxin-conjugated anti-p75NTR antibody.

## 1. Introduction

Neuroblastoma, the most common extracranial solid tumor in infancy, arises from the developing sympathetic nervous system [1,2]. Clinically, neuroblastoma is a heterogeneous disease with some forms regressing spontaneously and others progressing with high morbidity and mortality. The high-risk forms of the disease are treated with intensive therapy, including surgery, myeloablative chemotherapy, radiation, and immunotherapy. However, a large percentage of the high-risk patients have a poor clinical outcome and undergo relapses that are resistant to therapy [1,2]. Therefore, there is an urgent need for identifying new therapeutic strategies to fight the most aggressive forms of the disease.

Histone deacetylase (HDAC) inhibitors are relatively new anticancer drugs that have proven efficacy against highly malignant tumors, including neuroblastoma [3,4,5,6]. These drugs promote the hyperacetylation of histones and non-histone proteins, such as transcription factors, tubulin, and heat shock proteins [7]. By affecting epigenetic and non-epigenetic regulatory pathways, HDAC inhibitors have been found to induce differentiation, apoptotic death, and cell cycle arrest of cancer cells [3,4,5,6,7].

Neurotrophins and their receptors are known to play an important role in controlling the growth and progression of neuroblastoma [8,9,10] and interventions aiming at modifying their activity may have therapeutic utility in this disease [11,12,13,14]. We have recently reported that in human neuroblastoma cells valproic acid (VPA), a preferential inhibitor of HDACs belonging to class I and IIa [3], and other HDAC inhibitors favorably alter neurotrophin receptor expression and signaling. Thus, the exposure to HDAC inhibitors caused the downregulation of TrkB expression and signaling [15], which promote neuroblastoma cell survival and aggressiveness [9,10], and the upregulation of the truncated isoform of TrkC that is associated with neurotrophin 3-induced apoptosis [16]. Moreover, the treatment of neuroblastoma cells with HDAC inhibitors upregulated the expression of the common neurotrophin receptor p75NTR and the co-receptor sortilin, thus allowing the execution of proNGF-induced cell death [17]. The induction of p75NTR expression by HDAC inhibitors has been shown to occur through an epigenetic mechanism involving depletion of the methyltransferase EZH2, a core component of the polycomb repressor complex 2, and the consequent derepression of the transcription factor CASZ1, a positive regulator of the NGFR gene, encoding for p75NTR [17,18,19]. In addition, it has been reported that HDAC1 inhibition impairs the activity of a SP1/MIZ1/MYCN repression complex, which negatively control the NGFR promoter [20].

The efficient upregulation of p75NTR that is elicited by HDAC inhibitors suggested the possibility of employing this receptor as a cell surface target of the cytotoxic activity of specific antibody-drug conjugates. Preclinical studies and clinical trials have shown that immunotoxins containing the ribosome-inactivating protein saporin-S6 can be valid resources in immunotherapy of haematological and solid tumors [21,22,23]. Saporin-S6-containing immunotoxins are thought to be internalized by receptor-mediated endocytosis and degraded in the lysosome with the consequent release of saporin-S6, which is resistant to proteases [24]. Once inside the cell, saporin-S6 has been found to exert cytotoxic effects by inducing different death pathways, including apoptosis, necroptosis, autophagy, and irreversible inhibition of protein synthesis [21]. Clinical studies have shown that immunotoxins targeting cell surface molecules can show efficacy in tumors refractory to conventional therapy [21].

In the present study we investigated whether the upregulation of p75NTR elicited by HDAC inhibitors renders human neuroblastoma cells sensitive to the cytotoxic action of an anti-p75NTR antibody conjugated to saporin-S6 (p75IgG-Sap).

## 2. Results

### 2.1. Upregulation of p75NTR by Valproic Acid and Entinostat in Different Human Neuroblastoma Cell Lines

We first examined the induction of p75NTR by two well-characterized HDAC inhibitors, VPA and entinostat in different neuroblastoma cell lines. Entinostat is a selective class I HDAC inhibitor [25] and, similar to VPA [6,7], has been investigated as a potential anticancer agent in haematological and solid tumors [7,26]. The panel of human neuroblastoma cell lines that were examined included cells without (SH-SY5Y) and with (LAN-1, Kelly, BE(2)C, IMR-32, and NB-1) amplification of the *NMYC* gene, which is commonly associated with advanced stage tumors and considered a poor prognostic factor in neuroblastoma [1,2]. Moreover, the cell lines differed with regard to the genomic status of the oncogene anaplastic lymphoma kinase (*ALK*), a driver of neuroblastoma growth [1,2]. NB-1 cells are known to exhibit *ALK* amplification, whereas SH-SY5Y, LAN-1, and Kelly cells harbor the gain-of-function point mutation F1174L. As shown in Figure 1A, in all neuroblastoma cell lines that were examined prolonged exposure (24 h) to either VPA (1 mM) or entinostat (1 µM) increased p75NTR protein levels by several fold (SH-SY5Y: 14.0- and 18.0-fold; LAN-1: 3.9- and 10.0-fold; Kelly: 15.0- and 34.0-fold; IMR-32: 14.8- and 25.0-fold; BE(2)C: 16.7- and 36.0-fold, NB-1: 9.0- and 45.0-fold by VPA and entinostat, respectively; *p* < 0.001, *n* = 4). The exposure to the HDAC inhibitors increased the expression of the mature p75NTR protein of ~75 kDa and the ~60 kDa isoform corresponding to the less glycosylation state of the receptor [27]. In each neuroblastoma cell line that was examined, entinostat displayed a greater efficacy than VPA (*p* < 0.05, *n* = 4). Entinostat was also found to be more effective than VPA in increasing the levels of the p75NTR inducer CASZ1 in SH-SY5Y and LAN-1 cells (Figure 1B).

The concentration-response experiments that were performed in SH-SY5Y and Kelly cells showed that entinostat induced a significant increase of p75NTR protein levels at a concentration as low as 0.1 µM, with maximal effects at 1.0 and 3.0 µM and EC_50_ values of 0.55 ± 0.09 µM and 0.67 ± 0.05 µM, respectively (Figure 1C).

VPA (1 mM) and entinostat (1 µM) had either no effect or slightly increased p75NTR expression in non-neuroblastoma human cell lines, including human dermal fibroblasts (HDF), human embryonic kidney 293 cells (HEK 293), human prostate carcinoma PC-3 cells, and human cervical carcinoma HeLa cells (Appendix A).

### 2.2. Enhancement of p75NTR Cell Surface Expression by VPA and Entinostat

We next compared the efficacy of the two HDAC inhibitors in increasing the expression of p75NTR at the plasma membrane of neuroblastoma cells. The cell surface protein biotinylation experiments that were performed in SH-SY5Y cells that were pretreated for 24 h with either VPA (1 mM) or entinostat (1 µM) showed that both agents markedly increased the plasma membrane expression of p75NTR (Figure 2A). However, the increase elicited by entinostat was almost double that of VPA (*p* < 0.01, *n* = 3). Thus, in subsequent experiments entinostat was used as inducer of p75NTR expression.

Immunofluorescence analysis of the plasma membrane p75NTR using non permeabilized SH-SY5Y cells and an ATTO-488-conjugated anti-p75NTR antibody recognizing the extracellular domain of the receptor indicated that the exposure to entinostat increased the percentage of positive cells from 7.5 ± 4.2 to 72.8 ± 9% (*p* < 0.001) (Figure 2B). The receptor labeling appeared to be predominantly localized at the cell periphery and was abrogated when cells were pretreated with an excess of an antibody that was directed against the extracellular domain of p75NTR (p75-IgG) (Figure 2B), thus supporting the specificity of the immunostaining.

### 2.3. Internalization of p75NTR and Intracellular Localization of Saporin-S6 following Exposure to p75IgG-Sap

We then asked whether the enhanced cell surface expression of p75NTR that was induced by entinostat facilitated the uptake of p75IgG-Sap by neuroblastoma cells. Western blot experiments showed that the immunotoxin recognized p75NTR protein in SH-SY5Y cell lysates, whereas a control non-targeting preimmune antibody conjugated to saporin-S6 (IgG-Sap) detected no immunoreactive bands (Figure 3A). To assess whether the binding of p75IgG-Sap promoted the internalization of p75NTR, we examined the effects of the immunotoxin on the plasma membrane levels of the receptor that were determined by cell surface protein biotinylation. The SH-SY5Y cells were preincubated for 24 h with either vehicle or entinostat (0.3 and 1.0 µM) and then treated with vehicle or p75IgG-Sap (30 nM) for additional 24 h. As shown in Figure 3B, the exposure to the immunotoxin caused a marked decrease in the cell surface expression of p75NTR that was induced by each entinostat concentration. Additionally, double immunofluorescence analysis of p75NTR and lysosome-associated membrane protein 1 (LAMP1), a selective marker of late endosomes and lysosomes, showed that in SH-SY5Y cells that were pretreated with entinostat and exposed to the immunotoxin, p75NTR labeling displayed a diffused dotted pattern, which was partly (20.3 ± 7.7%) colocalized with LAMP1 immunoreactivity (Figure 3C). Conversely, in cells that were pretreated with entinostat and subsequently exposed to control IgG-Sap conjugate (30 nM), p75NTR labeling showed a patchy distribution along neurites and around the cell bodies and little colocalization with LAMP1 immunoreactivity (2.2 ± 0.6%) (Figure 3C).

The cellular distribution of saporin-S6 following cell exposure to p75IgG-Sap was investigated by using a rabbit polyclonal antibody, which in Western blots were found to recognize a major immunoreactive band of about 30 kDa corresponding to the molecular mass of saporin-S6 (Appendix A). Saporin-S6 immunofluorescence was scarcely detectable in vehicle-pretreated cells, but clearly evident in entinostat-pretreated cells, where it mostly appeared as a granular staining that was distributed in the close proximity to the nuclei (Figure 3D). Quantification of saporin-S6 immunoreactivity indicated that the entinostat treatment increased the percentage of positive cells by about ten-fold (*p* < 0.01, *n* = 3), as compared to vehicle-treated cells (Figure 3C). As observed for p75NTR, in the entinostat-treated cells, a fraction (69 ± 9%) of saporin-6 immunoreactivity was found to be colocalized with LAMP1 labeling (Figure 3D).

### 2.4. Entinostat Predisposes Neuroblastoma Cells to p75IgG-Sap Cytotoxicity

To assess whether the entinostat-induced increase in the intracellular localization of saporin-S6 translated into enhanced cytotoxicity, we examined the viability of SH-SY5Y cells that were preincubated for 24 h with either vehicle or 1 µM entinostat and then exposed for additional 24 h to either the vehicle or increasing concentrations of p75IgG-Sap. As shown in Figure 3E, propidium iodide staining to mark dead cells indicated that the entinostat treatment significantly decreased the cell viability. The subsequent addition of p75IgG-Sap had no effect in the vehicle-pretreated cells but enhanced the percentage of dead cells in entinostat-pretreated samples. The enhancement that was produced by p75IgG-Sap was concentration-dependent with an estimated EC_50_ value of 19 ± 5 nM (*n* = 5). Cell analysis by light microscopy showed that the exposure to p75IgG-Sap (30 nM) failed to affect the cell morphology when it was added after the vehicle, but markedly increased cell rounding, shrinking, and floating in entinostat-pretreated cells (Figure 3F). Similarly, by using a luminescence assay to measure the cell viability, we found that the addition of p75IgG-Sap (30 nM) caused a significant increase of cell death in entinostat- but not vehicle-pretreated cells (Figure 3G). Cell treatment with either IgG-Sap (30 nM) (Figure 3H) or unconjugated saporin-S6 (Sap) (Figure 3I) had no effects on the viability in either the vehicle- or entinostat-pretreated cells. To further examine the specificity of the cytotoxic effect that is elicited by p75IgG-Sap, the entinostat-pretreated cells were incubated with an excess of p75-IgG before the addition of p75IgG-Sap. As shown in Figure 3J, the incubation with unconjugated p75IgG (100 nM) completely prevented the cytotoxic effect of p75IgG-Sap.

### 2.5. Exposure to p75IgG-Sap Potentiates Entinostat-Induced Apoptosis of Neuroblastoma Cells

The accumulation of the cleaved active form of the caspase 3 is a hallmark of apoptotic cell death. Western blot experiments showed that the treatment of SH-SY5Y cells with 0.3 and 1 µM entinostat caused a concentration-dependent increase in the levels of cleaved caspase 3 (Figure 4A). The subsequent addition of p75IgG-Sap (30 nM) significantly enhanced the response at both entinostat concentrations, while having no effect in the cells that were pretreated with the vehicle (Figure 4A). The visualization of cleaved caspase 3 accumulation by immunofluorescence microscopy indicated that the sequential treatment with entinostat and p75IgG-Sap yielded a significant increase in the percentage of positive cells as compared to entinostat alone (Figure 4B). A greater stimulation of caspase 3/7 activity was also detected when the cells were treated with entinostat plus p75IgG-Sap (Figure 4C). Similar results were obtained in LAN-1 cells, where the addition of the anti-p75NTR immunotoxin (30 nM) enhanced both the formation of cleaved caspase 3 and the stimulation of caspase 3/7 activity that was elicited by entinostat (1 µM) (Figure 4D,E).

In the apoptotic cascade proteolytic cleavage of the DNA repairing enzyme poly (ADP-ribose) polymerase (PARP) by activated caspases 3 and 7 is a key event contributing to cell damage. Caspase-catalyzed PARP cleavage between Asp214 and Gly215 generates a 89 kDa fragment that accumulates in apoptotic cells. In line with caspase 3 activation, treatment of either SH-SY5Y or LAN-1 cells with entinostat increased the formation of the cleaved PARP fragment and this effect was potentiated by the subsequent addition of p75IgG-Sap (Figure 4F,G). Moreover, an in situ terminal transferase dUTP nick end-labeling fluorimetric assay to detect DNA fragmentation, another typical parameter of apoptosis, showed that there was a further increase in the percentage of cells displaying positive staining when p75IgG-Sap was added to cell pretreated with entinostat (Figure 4H).

Survivin, a member of the family of inhibitor of apoptosis proteins [28], is known to promote cell survival by binding several apoptosis-regulating factors leading to the inhibition of caspase enzymatic activity [29]. The treatment of SH-SY5Y cells with entinostat (1 µM) caused a decrease in survivin protein levels and this effect was potentiated by the subsequent exposure to p75IgG-Sap (30 nM) (Figure 4I). Conversely, the addition of the immunotoxin to vehicle-pretreated cells failed to affect the cellular levels of survivin.

### 2.6. Entinostat Upregulates p75NTR and Induces Apoptosis in Neuroblastoma Multicell Spheroids

In comparison to monolayer cell systems, three-dimensional cultures of cancer cells, such as spheroids, are considered as a more representative model of solid tumor where to study the biological effects of anticancer drugs [30]. We, therefore, investigated the effects of entinostat in multicell spheroids that were generated by either IMR-32 or SH-SY5Y cells. Spheroids were grown for 24 h and then treated for 72 h with either vehicle or 1 µM entinostat. As shown in Figure 5A,D, entinostat significantly inhibited the growth of spheroids that were obtained from either cell line. The reduction in spheroid size that was elicited by the HDAC inhibitor was associated with a marked increase in the expression of p75NTR protein (Figure 5B,E) and induction of apoptosis, as indicated by the increased formation of cleaved PARP (Figure 5C,F).

### 2.7. Treatment with p75IgG-Sap Enhances the Cytotoxic Effect of Entinostat in Neuroblastoma Multicell Spheroids

Having observed that entinostat was able to induce p75NTR expression in multicell spheroids, we investigated whether the exposure to p75IgG-Sap potentiated the antineuroblastoma activity of the HDAC inhibitor in this three-dimensional cell model. After 24 h of growth, SH-SY5Y spheroids were pretreated for 24 h with either vehicle or 1 µM entinostat and then incubated with p75IgG-Sap (30 nM). The spheroid size was then determined every 24 h up to 120 h. As shown in Figure 6A, the spheroids that were pretreated with vehicle and then exposed to p75IgG-Sap showed a growth rate that was not different from that of the spheroids that were treated with the vehicle alone. On the other hand, the addition of the anti-p75NTR immunotoxin to entinostat-pretreated spheroids caused a significantly greater inhibition of the tumor growth, as compared to that which was induced by entinostat alone.

We next investigated the effects of the sequential treatment with entinostat and p75IgG-Sap on tumor spheroids that were cultured for six days, that is when they reach a well-developed stage. SH-SY5Y spheroids that were pretreated for 48 h with either vehicle or 1 µM entinostat were then incubated for 72 h with either vehicle or 30 nM p75IgG-Sap. The spheroids that were treated with the vehicle showed an irregular morphology with protuberances of the outermost layer, which is the site of tumor growth [30] (Figure 6B). Entinostat-treated spheroids displayed a significantly smaller size and a more regular shape with few protuberances. The spheroids that were treated sequentially with entinostat and p75IgG-Sap had a round shape and a smaller size, as compared with those that were treated with the HDAC inhibitor alone (Figure 6B). The addition of p75IgG-Sap to the vehicle-pretreated spheroids did not cause significant morphological changes. Analysis of PARP cleavage indicated that the addition of the immunotoxin to entinostat-pretreated spheroids potentiated the stimulatory effect that was induced by the HDAC inhibitor, while having no significant effect on vehicle-pretreated tumor aggregates (Figure 6C).

### 2.8. Entinostat Upregulates p75NTR Expression in Neuroblastoma Xenografts

The in vivo ability of entinostat to increase p75NTR expression was examined in tumor xenografts that were established in athymic nude BALB/c mice by subcutaneous injection of SH-SY5Y cells. The mice bearing tumors with a volume of 0.67 ± 0.12 cm^3^ were treated daily for 10 days with either entinostat (20 mg/kg) or vehicle, both given by oral gavage. Western blot analysis showed that there was a five-fold increase (*p* < 0.001) in the protein levels of p75NTR in tumor xenografts of mice treated with entinostat as compared to the controls (Figure 7A). The in vivo treatment with entinostat also caused a significant increase in the levels of acetylated histone H3 in tumor xenografts (Figure 7B). On the other hand, entinostat-treated mice showed no significant changes in the expression of p75NTR protein in the liver, heart, kidney, and cerebellum as compared to the vehicle-treated mice (Figure 7C–F).

### 2.9. Antineuroblastoma Effect of Intratumoral Administration of p75IgG-Sap

We then examined the effect of the sequential treatment of the tumor-bearing nude mice with entinostat and p75IgG-Sap. In this experiment, the animals were pretreated for 10 days with either the vehicle or entinostat as described above, and then injected with either the vehicle or p75IgG-Sap (5.0 µg) into two distal sites of the tumor. The immunotoxin treatment was repeated after 48 h. Measurement of the tumor volumes at the end of the treatment showed a modest and non-significant reduction in mice that were treated with the test agents as compared to the control (vehicle plus vehicle) (Figure 8A). Upon resection, the tumors of entinostat plus p75IgG-Sap appeared friable, whereas those that were obtained from the other experimental groups were more compact (Figure 8B). Western blot analysis revealed a significant increase of cleaved PARP only in the lysates of tumors that were obtained from entinostat plus p75IgG-Sap-treated mice (Figure 8C). The determination of survivin protein levels showed a significant decrease in tumors of entinostat-treated animals and a more pronounced reduction in those of entinostat plus p75IgG-Sap-treated animals (Figure 8C).

To gain information about the tumor vitality, we examined the ability of tumor specimens to survive when they were cultured in vitro. Following 96 h of incubation, the majority of tumor fragments that were obtained from mice that were treated with either vehicle, p75IgG-Sap, or entinostat formed globular cell masses which remained firmly attached to the surface of the vessel upon medium change. These samples also showed numerous cells sprouting out of the masses. In contrast, fragments of tumors that were resected from entinostat plus p75IgG-Sap-treated mice showed little or no adhesion to the dish and easily detached upon medium renewal (Figure 8D).

## 3. Discussion

To selectively destroy cancer cells with the use of antibodies that were conjugated to cytotoxic drugs or toxins it is necessary that the target antigen is either tumor-specific or expressed in cancer cells at much higher levels than normal cells. In the present study we show that the exposure to the HDAC inhibitors entinostat and valproic acid markedly enhanced p75NTR expression in different human neuroblastoma cell lines in a manner that was apparently independent of *NMYC* and *ALK* genotypes. On the other hand, the HDAC inhibitors either failed to change or had minimal stimulatory effects on p75NTR expression in a panel of non-neuroblastoma cells, indicating that the strong p75NTR induction was not a generalized cellular response to the drugs but occurred in neuroblastoma cells with a certain degree of selectivity.

As compared to VPA, entinostat consistently behaved as a more efficient inducer of p75NTR levels that were measured in either whole cell lysates or cell surface protein preparations. Entinostat was also found to induce a greater enhancement of CASZ1 levels in SH-SY5Y, and LAN-1 cells. These findings suggest that the superior efficacy of entinostat in inducing p75NTR was the result of a greater action on the epigenetic machinery that controls the expression of the neurotrophin receptor in neuroblastoma cells [17,18,19]. The induction of p75NTR by entinostat occurred within a concentration range (0.1–1.0 µM) which correlated with the reported micromolar potency of the drug in inhibiting HDAC activity [31] and with the plasma concentrations that were required to produce anti-cancer effects in clinical trials [26].

For targeted immunotherapy, a crucial event is the receptor-mediated endocytosis of the immunotoxin conjugate. A line of evidence indicates that in entinostat-treated SH-SY5Y cells, upregulated p75NTR receptors underwent immunotoxin-dependent internalization and mediated the efficient intracellular delivery of saporin-S6. Thus, exposure to p75IgG-Sap caused a marked drop in the levels of plasma membrane p75NTR and promoted the receptor localization in LAMP1-positive cellular compartments, whereas treatment with control IgG-Sap failed to affect the receptor distribution. The addition of p75IgG-Sap yielded a greater accumulation of saporin-S6 immunoreactivity in entinostat-preteated cells as compared to vehicle-pretreated cells, in line with the marked difference in p75NTR expression levels. The finding that the addition of an excess of unconjugated p75-IgG before p75IgG-Sap markedly reduced the number saporin-S6-positive cells suggests that the toxin delivery by the immunotoxin was a receptor-dependent process. Moreover, the observations that in entinostat-pretreated cells that were exposed to p75IgG-Sap, the immunoreactivities of both saporin-S6 and p75NTR displayed granular staining patterns that were typical of endocytic vesicle and colocalization with LAMP1 labeling indicate that the immunotoxin was taken up by the cells through endocytosis.

As previously observed with VPA [16,17], prolonged treatment with entinostat caused a significant impairment of SH-SY5Y cell viability. This finding is in agreement with a previous study showing that entinostat exerted anti-tumor activity in various pediatric cancer cell lines, including the SMS-KCNR and SK-N-AS neuroblastoma cell lines [32]. We found that the cytotoxic effect of entinostat was markedly enhanced by the subsequent addition of nanomolar concentrations of p75IgG-Sap. The immunotoxin did not affect the viability of vehicle-pretreated cells, which is consistent with the poor intracellular delivery of saporin- S6 under this condition. On the other hand, in entinostat-pretreated cells the lack of effect by either unconjugated saporin-S6 or IgG-Sap and the blockade of the enhancement of cell death by an excess of p75IgG demonstrate the specificity and the receptor dependency of the immunotoxin action.

The cytotoxic activity of p75IgG-Sap treatment was further characterized by examining the influence on apoptosis, a major cell death pathway that is activated by HDAC inhibitors [3,4,5,6,7]. The exposure to the immunotoxin was found to enhance the expression of several key apoptotic processes that were stimulated by entinostat in neuroblastoma cells, including caspase 3 activation, PARP cleavage, and DNA fragmentation. These results indicate that, following intracellular delivery, saporin-S6 interacted positively with entinostat to trigger the apoptosis of neuroblastoma cells. The precise mechanisms by which these agents interact to induce programmed cell death have not been investigated. Nonetheless, it was found that treatment of SH-SY5Y cells with entinostat significantly reduced the levels of survivin, a crucial molecular brake on the caspase cascade. This finding is in agreement with previous studies showing that in different cancer cells HDAC inhibitors curtail survivin expression by multiple mechanisms, including reduction of transcription, altered post-translational modification, and increased proteosomal degradation [33,34,35]. Most importantly, the inhibitory effect of entinostat on survivin expression was enhanced by the subsequent exposure to p75IgG-Sap, possibly through saporin-S6-induced suppression of protein synthesis [36]. In neuroblastoma, high survivin levels are associated with recurrent disease and death [37,38,39,40], an observation that makes this protein a potential therapeutic target [41,42,43]. Thus, the enhanced inhibition of survivin steady-state levels that were obtained by the sequential treatment with entinostat and p75IgG-Sap may represent a novel strategy to impair the viability of survivin overexpressing cancer cells.

In addition to apoptosis, some HDAC inhibitors, including entinostat, have been shown to induce autophagy, an additional form of programmed cell death, in different cancer cells [44]. Autophagy has also been proposed to occur following intracellular delivery of saporin-S6 [22]. However, it remains to be investigated whether the exposure to entinostat or the combination of entinostat with the immunotoxin can trigger autophagic death in human neuroblastoma cells.

Three-dimensional cultures of cancer cells are considered advanced models that can fill the gap between traditional two-dimensional cultures and in vivo tumors [45]. By mimicking the intercellular interactions and the pathophysiological milieu of solid tumors, three-dimensional cell systems can more reliably predict the in vivo efficacy of anti-tumoral treatments, as compared to two-dimensional cultures [30,46]. Previous proteomic analysis has revealed important differences in protein expression between two- and three-dimensional cultures of human neuroblastoma cells [47]. In this regard, an important finding of the present study is that the exposure to entinostat retained the ability of inducing the upregulation of p75NTR expression in neuroblastoma multicell spheroids. Furthermore, entinostat inhibited the growth rate of developing spheroids and reduced the size and viability of mature spheroids. The effects of entinostat were potentiated by the subsequent addition of p75IgG-Sap, highlighting the anti-neuroblastoma effectiveness of the combination treatment in a three-dimensional cell system, which is known to display higher resistance to chemotherapeutics than two-dimensional cultures [48,49].

The in vivo administration of entinostat was found to upregulate p75NTR expression in tumor xenografts of SH-SY5Y cells that were developed in athymic nude mice. In the same tumors, there was an increase in histone H3 acetylation as compared to the controls, implying that the treatment was effective in inhibiting HDAC activity. Entinostat treatment failed to alter p75NTR protein levels in several mouse organs, such as heart, liver, kidney, and cerebellum, indicating that the drug preferentially targeted neuroblastoma tumors to induce p75NTR.

In line with the results that were obtained in vitro, the intratumoral administration of p75IgG-Sap had no effect in tumors that were pretreated with the vehicle but significantly induced apoptosis in those that were pretreated with entinostat, as shown by the increase in PARP cleavage and the decrease in survivin expression. Moreover, tumors of mice that were treated with entinostat plus immunotoxin displayed a low vitality when they were cultured in vitro, as indicated by their poor capacity to adhere to the substrate, further supporting the cytotoxic efficacy of the drug combination. However, at odds with the results that were obtained using multicell spheroids, neither the administration of entinostat nor the combined treatment with entinostat and p75IgG-Sap was found to induce a significant reduction in the volumes of the tumor xenografts, likely because of the high variability of this parameter within each experimental group and the limited duration of the treatment.

## 4. Materials and Methods

### 4.1. Materials

Entinostat (MS-275; SNDX 275) and VPA were purchased from Santa Cruz Biotechnology (Dallas, TX, USA) and Sigma-Aldrich (St. Louis, MO, USA), respectively. The immunotoxin p75IgG-Sap (ME20.4-SAP), non-targeted pre-immune mouse IgG antibody conjugated to saporin-S6 (IgG-Sap), and saporin-S6 that was purified from the seeds of *Saponaria officinalis* were obtained from Advanced Targeting Systems (San Diego, CA, USA).

### 4.2. Cell Culture

Human neuroblastoma cell lines SH-SY5Y, LAN-1, BE(2)-C, and IMR 32 were obtained from the European Collection of Authenticated Cell Cultures (ECACC) (Salisbury, UK), whereas the cell lines Kelly and NB-1 were from CLS Cell Lines Service GmbH (Eppelheim, Germany) and JCRB Cell Bank (Japan), respectively. The cell lines were authenticated by the vendors. SH-SY5Y, BE(2)-C, and LAN-1 cells were grown in Ham’s F12/MEM medium (1:1) (Sigma-Aldrich) containing 2 mM L-glutamine (Sigma-Aldrich) and 1% non-essential amino acids (NEAA) (Sigma-Aldrich). The IMR 32 cells were grown in MEM medium (Sigma-Aldrich) containing 2 mM L-glutamine and 1% NEAA. The Kelly and NB-1 cells were cultured in RPMI 1640 containing 2 mM L-glutamine (Sigma-Aldrich). The HeLa cells from human cervical carcinoma (ECACC) were grown in MEM medium that was supplemented with 2 mM L-glutamine, whereas human prostate carcinoma PC-3 cells (CLS Cell Lines Service GmbH) and human embryonic kidney 293 cells (HEK-293) (ECACC) were cultured in Ham’s F12/DMEM medium (1:1) containing 2 mM L-glutamine. The media were supplemented with 10% foetal calf serum (FCS) and 100 U/mL penicillin-100 µg/mL streptomycin (Sigma-Aldrich). Human dermal fibroblasts (HDF) that were isolated from adult skin were obtained from Life Technologies Co. (Carlsbad, CA, USA) and grown in medium 106 that was supplemented with Low Serum Growth supplement (Life Technologies Co.). The cells were maintained at 37 °C in a humidified atmosphere of 5% CO_2_ in air. Sub-confluent cultures were split every 72 h and seeded at the density of 1–3 × 10^4^/cm^2^ using 0.25% trypsin/EDTA (Sigma-Aldrich). After resuscitation, the cells were used for no more than 10–15 passages. The cells were periodically checked for mycoplasma contamination by using the MycoFluor Mycoplasma Detection kit (Invitrogen-Life Technologies).

### 4.3. Cell Treatment and Cell Lysate Preparation

Unless otherwise specified, neuroblastoma cells were washed with phosphate buffered saline (PBS) and incubated in medium containing 1% FCS. The cells were treated with the test agents as indicated in the text and maintained at 37 °C in a humidified atmosphere of 5% CO_2_ in air. The cell lysates were prepared by washing with PBS and scraping the cells into an ice-cold lysis buffer containing PBS, 0.1% sodium dodecyl sulphate (SDS), 1% Nonidet P-40, 0.5% sodium deoxycholate, 2 mM EDTA, 2 mM EGTA, 4 mM sodium pyrophosphate, 2 mM sodium orthovanadate, 10 mM sodium fluoride, 20 nM okadaic acid, 1 mM phenylmethylsulphonyl fluoride (PMSF), 0.5% phosphatase inhibitor cocktail 3, and 1% protease inhibitor cocktail (Sigma-Aldrich) (RIPA buffer). The samples were sonicated for 5 s in an ice-bath and aliquots of the cell extracts were taken for protein determination by the Bio-Rad protein assay (Bio-Rad Lab, Hercules, CA, USA).

### 4.4. Biotinylation of Cell Surface Proteins

Surface biotinylation of the cell proteins was performed as previously described [50]. Briefly, following treatment with the test agents, the SH-SY5Y cells were incubated for 45 min at 4 °C with the cell impermeable biotinylating agent sulfosuccinimidyl-6-(biotin-amido)hexanoate (0.50 mg/mL) (Pierce, Rockford, IL, USA). Thereafter, the cells were washed with PBS containing 20 mM glycine and solubilized by incubation in RIPA buffer that was supplemented with 1% Triton X 100. The cell extracts were centrifuged at 14,000× *g* for 5 min at 4 °C and the supernatants incubated overnight at 4 °C with streptavidin-conjugated agarose beads. Following washing, the beads were mixed with sample buffer and incubated for 2 min at 100 °C. The proteins were then analyzed by Western blot.

### 4.5. Western Blot Analysis

The cell proteins were separated by SDS-polyacrylamide gel electrophoresis (SDS-PAGE) and electrophoretically transferred to polyvinylidene difluoride membranes (Amersham Biosciences, Piscataway, NJ, USA) by using a semi-dry apparatus. Membranes were blocked with 5% low-fat dry milk, washed, and incubated overnight at 4 °C with one of the following primary antibodies: p75NTR (cat. no. 8238, Cell Signaling Technology, Danvers, MA, USA) (1:1000), CASZ1 (sc-398303, Santa Cruz Biotechnology, Dallas, TX, USA) (1:1000), pan-cadherin (cat. no. 4073, Cell Signaling Technology) (1:2000), poly (ADP-ribose) polymerase (PARP) (cat. no. 9542, Cell Signaling Technology) (1:1000), cleaved PARP (Asp214) (cat. no. 5625, Cell Signaling Technology) (1:1000); caspase 3 (cat. no. 9665, Cell Signaling Technology) (1:1000); cleaved caspase 3 (Asp175) (cat. no. 9664, Cell Signaling Technology) (1:1000); survivin (cat. no. 2808, Cell Signaling Technology) (1: 1000), histone H3 (acetyl-Lys9, Lys14) (cat. no. GTX122648, GeneTex Inc., Irvine, CA, USA) (1:1000), histone H3 (cat. no. GTX 122148, GeneTex Inc.) (1:2000), and actin (cat. no. A2066, Sigma-Aldrich) (1:2000). Thereafter, the membranes were washed and incubated with an appropriate horseradish peroxidase-conjugated secondary antibody (Santa Cruz Biotechnology). Immunoreactive bands were detected by using Clarity Western ECL substrate (Bio-Rad Lab.) and digital images were obtained by using either ECL Hyperfilm (Amersham) with Image Scanner III (GE Healthcare, Milan, Italy) or Luminescence Image analyzer LAS 4000 (FujiFilm, Tokyo, Japan). The band densities were determined using the NIH ImageJ software (US National Institutes of Health, Bethesda, MA, USA). For analysis of PARP and caspase 3, the formation of the cleaved form was normalized to the level of the corresponding uncleaved protein that was measured in the same sample. For the remaining immunoblots, the densitometric values were normalized to the levels of either actin or pan-cadherin, as indicated.

### 4.6. Immunofluorescence Analysis

Cells that were grown onto coverslips that were precoated with poly-L-lysine (Sigma-Aldrich) were exposed to the test agents as indicated in the text, washed, and fixed in 4% paraformaldehyde.

For analysis of p75NTR expression, nonpermeabilized cells were blocked with 3% BSA and incubated overnight with rabbit polyclonal ATTO-488-conjugated anti-p75NTR (extracellular) antibody (cat. no. ANT-007-AG, Alomone Labs, Jerusalem, Israel) (1:100). To assess the specificity of the labeling, the cells were preincubated for 2 h with an unlabeled mouse monoclonal antibody similarly directed against the extracellular domain of the receptor (sc-55467, Santa Cruz Biotechnology) (1:10).

For the detection of cleaved caspase 3, p75NTR, saporin-S6, and LAMP1 immunoreactivities, cells were permeabilized with 0.2% Triton X-100, blocked and incubated overnight with rabbit polyclonal anti-cleaved caspase 3 antibody (cat no. 9661, Cell Signaling Technology) (1:200), rabbit monoclonal anti-p75NTR antibody (cat. no. 8238, Cell Signaling Technology) (1:500), rabbit polyclonal anti-saporin antibody (cat. no. PA1-18425, Invitrogen) (1:200), and mouse monoclonal anti-LAMP1 antibody (sc-18821, Santa Cruz Biotechnology) (1:200). The control samples were incubated in the presence of preimmune IgG. Thereafter, the cells were incubated with the appropriate Alexa-Fluor488- or Alexa-Fluor594-conjugated secondary antibody (1:1500) (Invitrogen-Molecular Probes) and cell nuclei were stained with 0.1 µg/mL DAPI. The cells were analyzed with an Olympus BX61 microscope that was equipped with a F-View II CCD-camera and mirror units for the detection of green (U-MNIBA3), red (U-MNG2) and blue (U-MNUA2) fluorescence by using either 40X or 60X objective lens. Digital images were acquired using constant camera settings within each experiment and were analyzed using the program Cell P (Olympus Soft Imaging Solutions, Homburg, Germany). At least 10 fields were randomly selected for each sample and only cells showing an unobstructed nucleus or soma were considered.

For quantification, the average pixel intensity was measured within the region of the cell soma or the nucleus, as appropriate, and in an adjacent area, which was used as background value. The cells were deemed to be positive if the average pixel intensity was equal or above a threshold value corresponding to one standard deviation above the average pixel intensity of the respective control samples. No labeling was detected in the samples that were treated with preimmune IgG. The images were analyzed by an investigator unaware of the treatment.

### 4.7. Analysis of Living Cell Morphology

The cells were grown in 6-well plates and treated as specified in the text. The morphology of living cells was analyzed by phase-contrast light microscopy using an Olympus IX51 inverted microscope that was equipped with Plan achromatic objectives. The images were acquired in randomly selected fields by using an Olympus digital camera and analyzed by an investigator that was unaware of the treatment.

### 4.8. Assay of Cell Viability

For propidium iodide (PI) staining, the cells were grown on poly-L-lysine precoated glass coverslips and incubated with the test agents as indicated in the text. The cells were then exposed to 1 µg/mL PI (Sigma-Aldrich) for 1 h, washed, fixed in 4% paraformaldehyde, and incubated with 0.1 µg/mL DAPI. Coverslips were mounted with Fluoromount aqueous mounting medium (Sigma-Aldrich) and the cells were examined by fluorescence microscopy using an Olympus BX61 microscope. Images were captured over randomly selected fields and analyzed with the software Cell P. PI-positive cell nuclei were reported as percent of the total nuclei that were stained with DAPI. Three separate preparations were analyzed by an investigator that was unaware of the cell treatments.

Cell viability was also measured by luminescence analysis using the Real Time-GLO MT assay kit (Promega, Madison, WI, USA). The cells that were grown in 96-well plates (ViewPlate, PerkinElmer) were exposed to the test agents as indicated in the text and then incubated with the reagents that were provided by the kit following the manufacturer’s instructions. The luminescence intensity was measured by using a Wallac Victor III microplate reader (PerkinElmer). The assays were performed in triplicate.

### 4.9. Assay of Caspase Activity

The cells that were grown in 96-well plates (ViewPlate-96) were incubated as specified in the text. The cells were then assayed for caspase activity by using Caspase-Glo 3/7 assay kit (Promega), according to the manufacturer’s instructions. Luminescence intensity was measured by using a Wallac Victor III microplate reader. The assays were performed in triplicate.

### 4.10. Terminal Transferase dUTP Nick End-Labeling Assay (TUNEL Assay)

Cells that were grown on glass coverslips were incubated with the test agents for 24 h. In situ TUNEL assay was performed using the DeadEnd fluorimetric TUNEL system (Promega), according to the manufacturer’s instructions. The cell nuclei were stained with DAPI. The images were captured over randomly selected fields and analyzed with Cell P software. A total of three separate culture preparations were examined. The assays were performed in triplicate.

### 4.11. Tumour Spheroid Generation and Analysis

To generate spheroids, the cells were seeded in a 96-well U-bottom ultra-low attachment plate (Nunclon Sphera, Thermo Fisher Scientific) at the density of 3 × 10^3^ cells/well in 200 µL of complete growth medium. The plates were centrifuged at 200× *g* for 5 min at room temperature to generate one spheroid in each individual well and placed in an incubator at 37 °C and 5% CO_2_. The spheroids were incubated with the test agents as specified in the text 24 h after cell seeding. The spheroids were examined by light microscopy using an Olympus IX51 inverted microscope that was equipped with a 10X Plan achromatic objective and an Olympus digital camera. The images were acquired at the time points indicated in the text and the area of each spheroid was measured by using the ImageJ software and calibrated by using an image of known scale [46]. The assays were performed in quadruplicate. Measurements were made by an investigator that was unaware of the experimental design. For Western blot analysis, four spheroids from the same experimental group were pooled, centrifuged, resuspended in ice-cold RIPA buffer, and lysed by sonication. Aliquots of the lysates containing an equal amount of protein were mixed with sample buffer, heated at 100 °C, and subjected to SDS-PAGE.

### 4.12. Neuroblastoma Cell Xenografts

The experiments were conducted according to the European legislation (EU Directive 2010/63) and approved by the Animal Ethics Committee of the University of Cagliari and Italian Ministry of Health (auth. 759/2020/PR). The experimental protocols were designed to minimize pain and suffering and reduce the number of animals that were used. Female BALB/c athymic nude mice aged 5–6 weeks that were obtained from Charles River Laboratories (Calco, Italy) were housed in groups of four per cage under humidity- and temperature- (23–25 °C) controlled conditions and a light/dark cycle set at 12 h intervals. The animals were preserved under sterile conditions and fed sterilized food and water ad libitum. All the procedures were conducted under aseptic conditions.

The SH-SY5Y cells were grown to 80% confluency, harvested, and suspended in a solution of Matrigel (BD Bioscience, Heidelberg, Germany)/PBS (1:1 *v*/*v*) at the concentration of 1 × 10^8^ cells/mL. The mice were anaesthetized and inoculated subcutaneously in the flank with 200 µL of cell suspension. The tumor volumes were assessed every other day by a digital calliper and calculated according to the formula: length × width^2^/2, where length > width. In a first experiment, the mice bearing mean tumor volumes of 0.67 ± 0.12 cm^3^ were randomly distributed in two groups (5 mice/group) and treated daily for 10 days with either entinostat (20 mg/kg) or vehicle (10% dimethyl sulfoxide plus 90% corn oil, Sigma-Aldrich) by oral gavage (0.1 mL/10 g). The dose of entinostat was chosen on the basis of a previous study showing that the chronic oral administration of the drug to nude mice produced no signs of toxicity up to 24.5 mg/kg [29]. A total of 24 h after the last treatment, the mice were sacrificed and tumors along with liver, heart, kidney, and cerebellum, were rapidly resected, snap frozen in liquid nitrogen, and stored at −80 °C for subsequent Western blot analysis.

In a second experiment, the mice with tumor volumes of 0.54 ± 0.09 cm^3^ were randomly distributed into two groups and pretreated with either vehicle or entinostat as described above. Thereafter, each group was subjected to intratumoral injection (50 µL) of either saline or p75IgG-Sap (5 µg) performed in two distant sites of the tumor. The injections were repeated after 48h. A total of 48 h after the last injection, the mice were sacrificed and tumors were resected under aseptic conditions. One portion of each tumor was rapidly frozen and stored at -80 °C for Western blot analysis, while the other portion was minced with a scalpel into ~1 mm^3^ fragments. A total of 12 fragments of each tumor were seeded into individual plastic dishes containing complete culture medium. The dishes were incubated at 37 °C under CO_2_. After 96 h, the medium was changed and the cultures were analyzed by light microscopy to examine the growth and determine the fraction of tumor fragments which remained adherent to the vessel.

### 4.13. Statistical Analysis

The results are reported as the mean ± SD. Statistical analysis was performed by using the program Graph Pad Prism (San Diego, CA, USA). The EC_50_ values were determined by nonlinear regression curve-fitting of the concentration-response data. The data are expressed as a percentage or fold stimulation of the control, which was included in each independent experiment. The control group was set as 100 or 1 with a variance obtained by expressing each control value as a percentage of the mean of the raw values of the control group. In the experiments where the control values were equal to zero, values of experimental groups were expressed as a percentage of the maximal effect set as 100. The variance of this value was determined in the same manner as for the control group. Unpaired Student’s *t*-tests or analysis of variance (ANOVA) followed by Tukey’s or Bonferroni’s tests was performed, as appropriate, to assess significant differences between the experimental groups. A value of *p* < 0.05 was considered to be statistically significant.

## 5. Conclusions

Previous investigations have shown that ectopic expression of p75NTR triggers apoptotic death of human neuroblastoma cells, suggesting that p75NTR may function as a tumor suppressor gene in this disease [51,52,53]. Moreover, clinical studies have reported that in primary neuroblastic tumors, p75NTR mRNA levels correlated with enhanced event-free and overall survival and suggested that induction of p75NTR expression could be an option to reduce tumorigenicity of neuroblastoma [54]. In this context, the present demonstration that entinostat is able to induce a strong upregulation of p75NTR constitutes an important finding supporting the use of this HDAC inhibitor as an antineuroblastoma agent. Our study shows that the p75NTR induction occurs predominantly in neuroblastoma cells and sensitizes the cells to the cytotoxic action of p75IgG-Sap, which potentiates the antineuroblastoma activity of entinostat both in vitro and in vivo models. Although further research is necessary, the findings provide preclinical evidence for developing a novel immunotherapeutic approach for neuroblastoma based on p75NTR induction in tumor cells and targeting of the upregulated receptor with antibodies conjugated to either toxins or cytotoxic drugs.

## Figures and Tables

**Figure 1 ijms-23-03849-f001:**
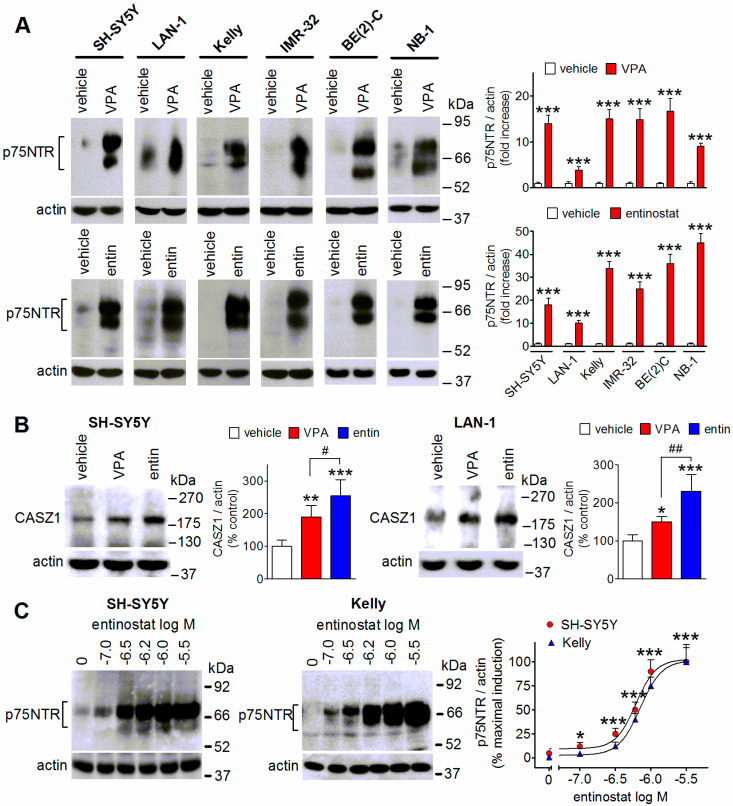
Induction of p75NTR expression in human neuroblastoma cell lines by VPA and entinostat. (**A**) The cells were exposed for 24 h to either vehicle, 1 mM VPA, or 1 µM entinostat (entin) and then analyzed for p75NTR protein levels by Western blot. The values are the mean ± SD of four independent experiments. The position of molecular mass standards is reported in the right side of the blots. *** *p* < 0.001 vs. control (vehicle) by Student’s *t* test. (**B**) SH-SY5Y and LAN-1 cells were treated as indicated in (**A**) and then analyzed for CASZ1 protein levels. The values are the mean ± SD of four (SH-SY5Y) and three (LAN-1) experiments. * *p* < 0.05, ** *p* < 0.01, *** *p* < 0.001 vs. control (vehicle). ^#^ *p* < 0.05, ^##^ *p* < 0.01 vs. control (vehicle) by ANOVA followed by Tukey’s test. (**C**) SH-SY5Y and Kelly cells were incubated for 24 h with either vehicle or the indicated concentrations of entinostat. The cell lysates were analyzed for p75NTR protein expression. Values are the mean ± SD of three experiments. * *p* < 0.05, *** *p* < 0.001 vs. control (vehicle) by Student’s *t* test.

**Figure 2 ijms-23-03849-f002:**
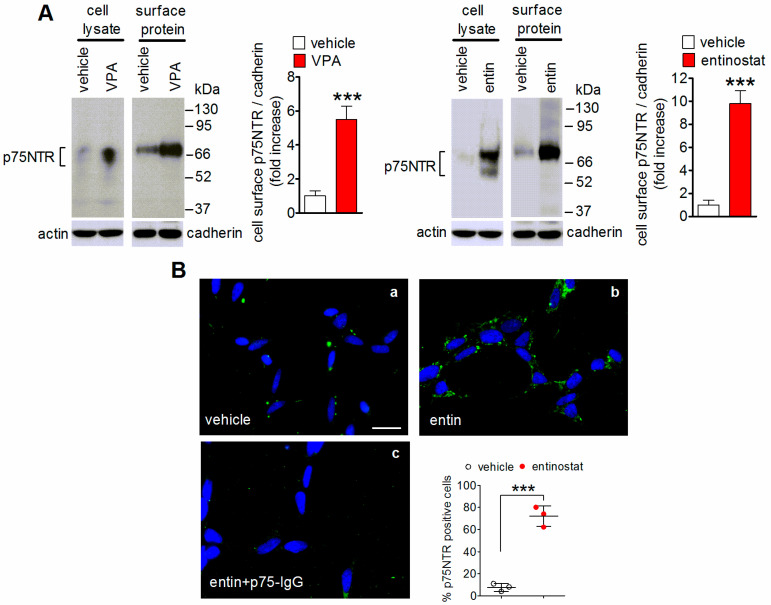
Enhancement of cell surface p75NTR expression by VPA and entinostat. (**A**) The SH-SY5Y cells were incubated for 24 h with either vehicle, 1 mM VPA, or 1 µM entinostat. Thereafter, the cells were treated with the cell impermeant biotinylating agent sulfosuccinimidyl-6-(biotin-amido) hexanoate and the solubilized proteins were isolated by precipitation with streptavidin-conjugated agarose beads. The total cell extract (cell lysate) and precipitated proteins (surface protein) were analyzed for p75NTR by Western blot. The bar graphs report the changes in the cell surface p75NTR levels that were normalized normalized to pan-cadherin (cadherin), a plasma membrane marker. Values are the mean ± SD of three independent experiments. *** *p* < 0.001 vs. vehicle by Student’s *t*-test. (**B**) SH-SY5Y cells that were grown on glass coverslips were treated for 24 h with either vehicle (**a**) or 1 µM entinostat (entin) (**b**,**c**), fixed, and incubated overnight with ATTO-488-conjugated anti-p75NTR (extracellular) antibody. In (**c**) the cells were preincubated with a mouse monoclonal unconjugated antibody directed against the extracellular domain of the receptor before the addition of the labeled antibody. The images were analyzed for p75NTR expression (green color) by fluorescence microscopy. The nuclei were stained in blue with 4′-6-diamidino-2phenylindole dihydrochloride (DAPI). Scale bar = 25 µm. The values that are shown in the scatter plot are the mean ± SD of three independent experiments. *** *p* < 0.001 vs. vehicle by Student’s *t*-test.

**Figure 3 ijms-23-03849-f003:**
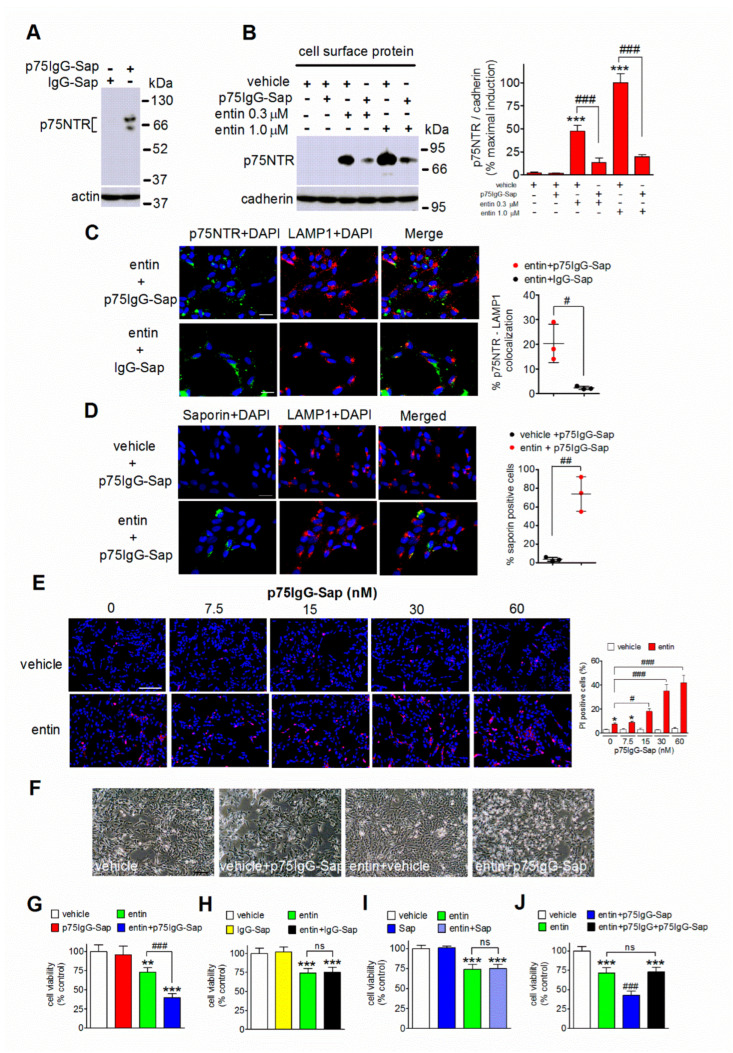
Exposure to p75IgG-Sap induces p75NTR internalization, intracellular saporin-S6 delivery, and citotoxicity in entinostat-treated SH-SY5Y neuroblastoma cells. (**A**) The cells were treated for 24 h with 1 µM entinostat and the cell lysates were analyzed for p75NTR immunoreactivity by Western blot using either a non-targeted preimmune antibody conjugated to saporin-S6 (IgG-Sap) (1:50) or p75IgG-Sap (1:50). The images are representative of three separate experiments. (**B**). The cells were incubated for 24 h with either vehicle or the indicated concentrations of entinostat (entin) and then exposed for additional 24 h with either vehicle or p75IgG-Sap (30 nM). Biotinylated cell surface proteins were isolated and analyzed for p75NTR expression. The values are the mean ± SD of three independent experiments. *** *p* < 0.001 vs. vehicle; ^###^, *p* < 0.001 by ANOVA followed by by Tukey’s test. (**C**) The cells were treated with 1 µM entinostat for 24 h and then exposed to either p75IgG-Sap (30 nM) or IgG-Sap (30 nM) for additional 24 h. The cells were then analyzed for p75NTR (green color) and LAMP1 (red color) localization by immunofluorescence. (**D**) The cells were incubated for 24 h with either vehicle or entinostat (1 µM), exposed to p75IgG-Sap (30 nM) for additional 24 h, and then analyzed for saporin-S6 (green color) and LAMP1 (red color) expression by immunofluorescence. In (**C**,**D**) the cell nuclei were stained in blue with DAPI. Scale bar = 25 µm. Values that are reported in the scatter plots are the mean ± SD of three independent experiments ^#^ *p* < 0.05; ^##^ *p* < 0.01 by Student’s *t*-test. (**E**) The cells were treated for 24 h with either vehicle or 1 µM entinostat and subsequently exposed to the indicated concentrations of p75IgG-Sap for 24 h. Cell viability was assayed by propidium iodide (PI) staining (red color). The values are the mean ± SD of five independent experiments. * *p* < 0.05 vs. vehicle; ^#^ *p* < 0.05, ^###^ *p* < 0.001 by ANOVA followed by Tukey’s test. (**F**) The cells were treated for 24 h with either vehicle or 1 µM entinostat and then exposed to either vehicle or 30 nM p75IgG-Sap. The images were obtained by phase-contrast microscopy and are representative of five separate experiments. (**G**–**J**) Cell viability was determined by a luminescence assay; (**G**) The cells were treated as indicated in (**F**); (**H**,**I**) The cells were treated with either vehicle or 1 µM entinostat and then incubated for 24 h with vehicle, 30 nM IgG-Sap, or 30 nM saporin-S6 (Sap); (**J**) The cells were treated for 24 h with either vehicle or 1 µM entinostat and then incubated with 30 nM p75IgG-Sap in the absence and in the presence of 100 nM p75IgG added 2 h before the immunotoxin. The values are expressed as percent of control (vehicle) the mean ± SD of four independent experiments. ** *p* < 0.01, *** *p* < 0.001 vs. vehicle; ^###^ *p* < 0.001, ns = not significant by ANOVA followed by Tukey’s test.

**Figure 4 ijms-23-03849-f004:**
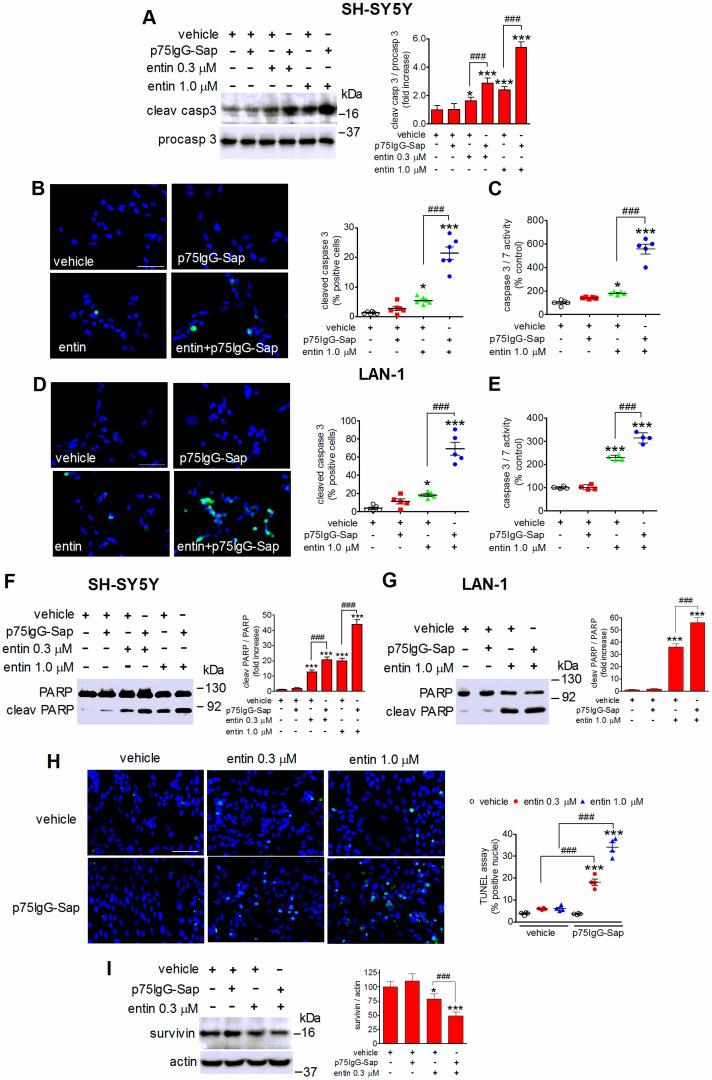
The immunotoxin p75IgG-Sap potentiates entinostat-induced apoptosis of human neuroblastoma cells. (**A**) SH-SY5Y cells were treated for 24 h with either vehicle or the indicated concentrations of entinostat (entin) and then exposed to either vehicle or 30 nM p75IgG-Sap for 24 h. The cell lysates were analyzed for cleaved and procaspase 3 levels by Western blot. (**B**,**D**) SH-SY5Y (**B**) and LAN-1 (**D**) cells were treated for 24 h with either vehicle or 1 µM entinostat and subsequently incubated with 30 nM p75IgG-Sap. The cells were analyzed for cleaved caspase 3 expression (green color) by immunofluorescence microscopy. The nuclei were stained in blue with DAPI. Scale bar = 50 µm. (**C**,**E**) SH-SY5Y and LAN-1 cells were treated as in (**B**,**D**) and the caspase 3/7 activity was determined by a luminescence assay. (**F**,**G**) SH-SY5Y and LAN-1 cells were pretreated for 24 h with either the vehicle or the indicated concentrations of entinostat and then exposed for additional 24 h to either the vehicle or 30 nM p75IgG-Sap. The cell lysates were analyzed for PARP cleavage. (**H**) SH-SY5Y cells were treated as in (**A**) and then analyzed for DNA fragmentation by using an in situ fluorimetric TUNEL assay. Scale bar = 50 µm. (**I**) SH-SY5Y cells were treated for 24 h with either the vehicle or 0.3 µM entinostat and then exposed for 24 h to either the vehicle or 30 nM p75IgG-Sap. The cell lysates were analyzed for survivin levels. Values are the mean ± SD of four independent experiments. * *p* < 0.05, *** *p* < 0.001 vs. vehicle; ^###^ *p* < 0.001 by ANOVA followed by Tukey’s test.

**Figure 5 ijms-23-03849-f005:**
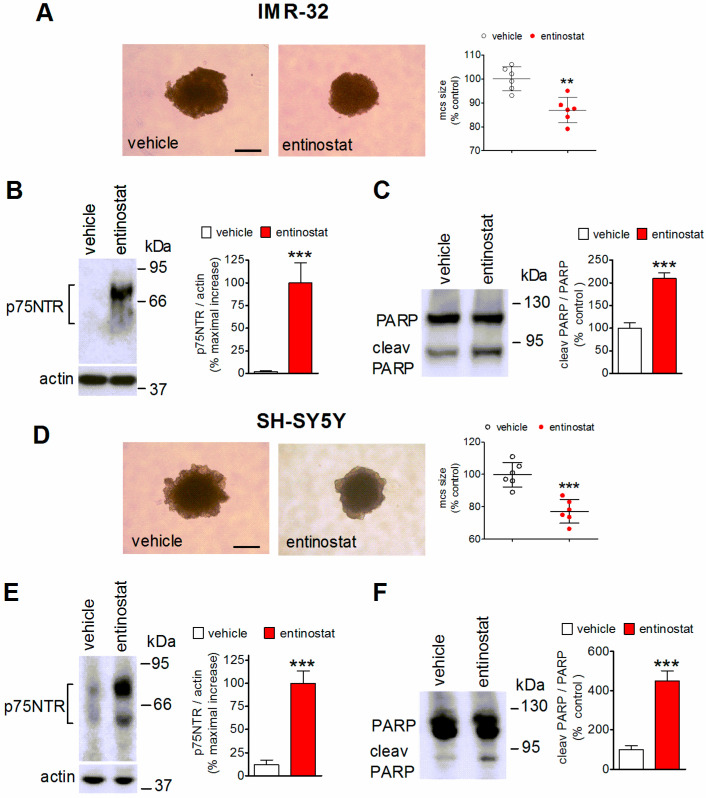
Upregulation of p75NTR and induction of apoptosis by entinostat in neuroblastoma spheroids. (**A**,**D**) Light microscopy images of IMR-32 and SH-SY5Y multicell spheroids (mcs) that were incubated for 72 h with either vehicle or 1 µM entinostat. The scatter plots report the values of mcs sizes expressed as percent of control (vehicle). Scale bar = 200 µm. (**B**,**C**,**E**,**F**) The spheroids were treated as indicated in (**A**,**D**) and then analyzed for p75NTR levels and PARP cleavage by Western blot. The values are the mean ± SD of four individual experiments. ** *p* < 0.01, *** *p* < 0.001 vs. vehicle by Student’s *t*-test.

**Figure 6 ijms-23-03849-f006:**
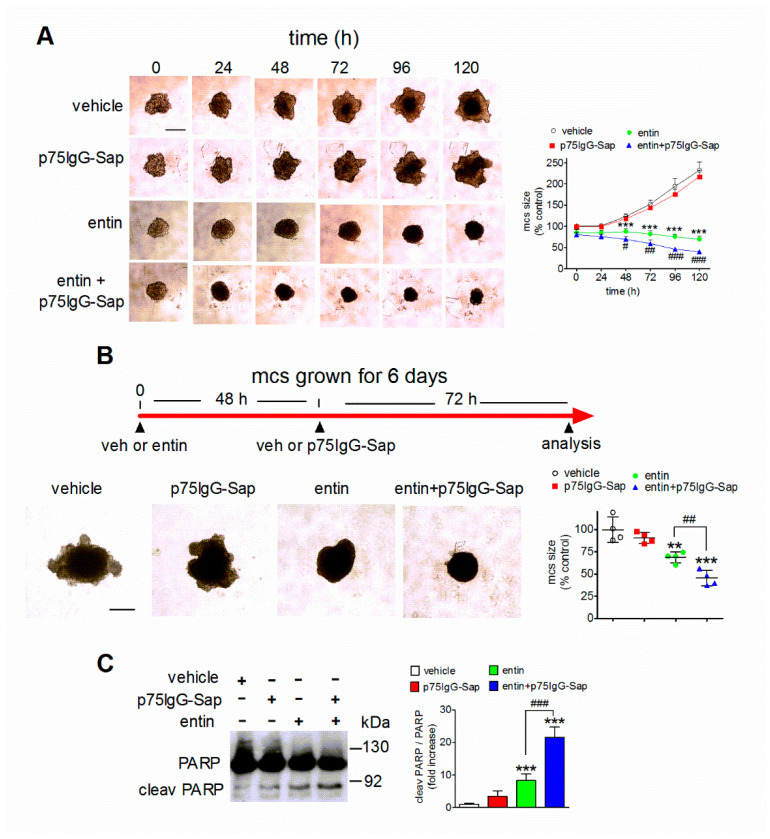
The combination treatment with p75IgG-Sap enhances entinostat cytotoxicity in neuroblastoma multicell spheroids. (**A**) A total of 24 h after cell seeding, SH-SY5Y multicell spheroids (mcs) were preincubated for 24 h with either vehicle or 1 µM entinostat (entin). Thereafter (time 0), the spheroids were incubated for the indicated time periods with either vehicle, 30 nM p75IgG-Sap, 1 µM entinostat, or the combination of entinostat plus p75IgG-Sap. Digital images were acquired by light microscopy at the indicated time points. The mcs size of each experimental group is reported as percent of the control (vehicle at time 0). Scale bar = 200 µm. Values are the mean ± SD of four independent experiments. *** *p* < 0.001 vs. control; ^#^ *p* < 0.05, ^##^ *p* < 0.01, ^###^ *p* < 0.001 vs. entinostat alone by ANOVA followed by Bonferroni’s test. (**B**) SH-SY5Y spheroids that were grown for six days were pretreated for 48 h with either vehicle or 1 µM entinostat and the exposed for 72 h to either the vehicle or 30 nM p75IgG-Sap. At the end of the incubation, digital images were acquired by light microscopy. The mcs size of each experimental group is reported as percent of the control (vehicle + vehicle). Scale bar = 200 µm. (**C**) SH-SY5Y spheroids that were grown for six days and treated as in (**B**) were analyzed for cleaved PARP by Western blot. Values are the mean ± SD of four independent experiments. ** *p* < 0.01, *** *p* < 0.001 vs. vehicle. ^##^ *p* < 0.01, ^###^ *p* < 0.001 by ANOVA followed by Tukey’s test.

**Figure 7 ijms-23-03849-f007:**
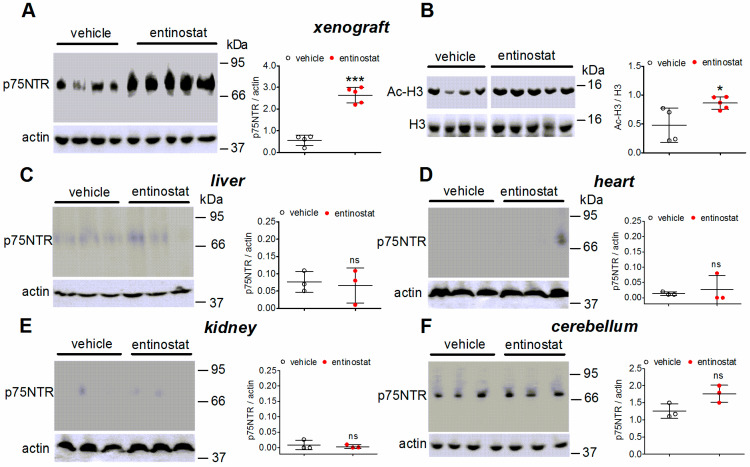
Expression of p75NTR and histone H3 acetylation in neuroblastoma tumor xenografts and different organs of entinostat-treated mice. (**A**,**C**–**F**) Athymic nude mice bearing xenograft of SH-SY5Y cells were treated daily for 10 days with either the vehicle or entinostat (20 mg/kg) by oral gavage. The mice were sacrificed, the tumor xenografts and the indicated organs were resected, and the tissue extracts were analyzed for p75NTR levels by Western blot. (**B**) The levels of acetylated histone H3 were determined in neuroblastoma tumor xenografts of vehicle- and entinostat-treated mice by Western blot. Each lane was loaded with a sample that was obtained from an individual animal (in (**A**,**B**): four animals that were treated with vehicle and five animals that were treated with entinostat; in (**C**–**F**), three animals that were treated with either vehicle or entinostat). Scatter plots indicate the absolute values of densitometric ratios and are the mean ± SD of three independent Western blots. * *p* < 0.05, *** *p* < 0.001, ns = not significant by Student’s *t* test.

**Figure 8 ijms-23-03849-f008:**
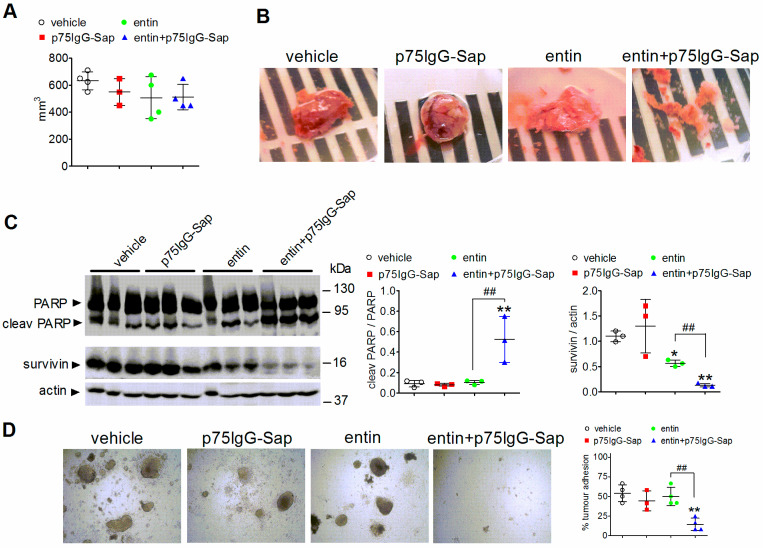
Cytotoxic effects of p75IgG-Sap in entinostat-treated neuroblastoma tumor xenografts. (**A**) Nude mice bearing xenografts of SH-SY5Y cells were treated for 10 days with either the vehicle or entinostat as indicated in Figure 7, and then injected twice with either the vehicle or p75IgG-Sap (5.0 µg) into two distal sites of the tumor. The scatter plot indicates the values (mean ± SD) of the tumor volumes that were measured 48 h after the last intratumoral injection. (**B**) Representative images of tumor xenografts that were acquired following resection from each experimental group. (**C**) Tumor xenografts of mice that were treated as indicated in (**A**) were analyzed for PARP cleavage and survivin expression by Western blot. Each lane was loaded with a sample th acquired at was obtained from an individual animal. Scatter plots indicate the absolute values of densitometric ratios and are the mean ± SD. (**D**) Tumor xenograft fragments were incubated for 96 h in complete growth medium. Thereafter, the medium was changed and the cell aggregates were analyzed by light microscopy to examine the growth and adhesion to the substrate. The scatter plot indicates the percent of tumor adhesion (mean ± SD) for each experimental group. * *p* < 0.05, ** *p* < 0.01 vs. vehicle; ^##^ *p* < 0.01 by ANOVA followed by Tukey’s test.

## Data Availability

Data are contained in the article and Appendix A.

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
