# Peer review of "Upregulation of p75NTR by Histone Deacetylase Inhibitors Sensitizes Human Neuroblastoma Cells to Targeted Immunotoxin-Induced Apoptosis"

_ijms, 2022, doi:10.3390/ijms23073849_

Round 1
Reviewer 1 Report
The manuscript entitled “Upregulation of p75NTR by histone deacetylase inhibitors sensitizes human neuroblastoma cells to targeted immunotoxin-induced apoptosis” describes the administration of p75IgG-Sap induced apoptosis only in tumours of mice pretreated with entinostat. These findings define a novel experimental strategy to selectively eliminate neuroblastoma cells based on the sequential treatment with entinostat and a toxin-conjugated anti-p75NTR antibody.
Overall, most of the results were supportive to the conclusions. The basic question is interesting and the data are consistent with most of the notion.
General Critique:
I would like to raise the following points:
- Line 93, in all neuro-93 blastoma cell lines examined prolonged exposure to either VPA or entinostat increased 94 p75NTR protein levels by several fold. Please clearly indicate increase ratio or how many fold?
- Line 361, Figure 7. One thing the authors need to address is that it is not clear how the western blot samples were used. There were three or five samples from three individual mice of each group? Were the samples pooled to give one band for each group? or each sample was run on separated western blot and then the results were averaged?
- Thus, the authors concluded that cytotoxic action of an anti-p75NTR antibody conjugated to the toxin saporin-S6 induced cell death in human neuroblastoma cells through only apoptosis and might be a potential novel complementary gene therapeutic agent for the treatment of neuroblastoma cancer cells. Is there other pathway either through autophagy or combined apoptosis and autophagy?
Author Response
point 1. As requested, fold increase of p75NTR levels by VPA and entinostat in each neuroblastoma cell line is now specified in the text (page 3, line 1).
point 2. The legend to Figure 7 has been changed to clearly indicate how the Western blot data were generated.
point 3. The Referee asked whether, in addition to apoptosis, other pathways of programmed cell death such as autophagy contributed to the cytotoxic actions of the immunotoxin. We have not performed experiments to investigate this issue. However, in the revised manuscript, we discussed this possibility (page 16). An new citation [Ref. 44] has been added to provide additional information to this point.
Reviewer 2 Report
The manuscript ‘Upregulation of p75NTR by histone deacetylase inhibitors sensitizes human neuroblastoma cells to targeted immunotoxin-induced apoptosis’ is interesting based on the therapeutic strategy in neuroblastoma. The manuscript is well written and experiments are included with proper controls.
Minor Comments:
- ‘As shown in Figure 1A, in all neuroblastoma cell lines examined prolonged exposure to either VPA or entinostat increased 94 p75NTR protein levels by several fold’ authors did not mention how long cells have been treated for Figure 1A and B.
- Does a combination of entin and p75IgG-Sap show synergy or additive effect?
Author Response
Point 1. The Referee requested to specify how long cells were treated in the experiments reported in Figures 1A and B. This information was provided in the legend to Figure 1 of the original manuscript and is now indicated in the text of the revised manuscript along with the concentration of VPA and entinostat used (Results 2.1);
Point 2. The Referee asked whether the combination of entinostat and p75IgG-Sap show synergy or additive effect. In our study the potentiation of entinostat cytotoxicity induced by p75IgG-Sap likely results from the increased intracellular delivery of saporin-S6 consequent to the upregulation of p75NTR. As discussed in the original manuscript (page 16), how entinostat and saporin-S6 interact to induce apoptosis has not been investigated. Therefore, we do not know whether the interaction is additive or synergistic. To make this point more clear, in page 16 the text was changed substituting “positively” for “synergistically” (page 16, line 470) and “enhanced” for “synergistic” (page 16, line 482).
It should be considered that in the present study p75IgG-Sap had no effect “per se” in vehicle- treated cells. Thus, experiments investigating whether, once inside the cell, saporin-S6 interacts additively or synergistically with entinostat should be performed by allowing the intracellular delivery of saporin-S6 independently of the anti-p75NTR antibody acting as a carrier (i.e. by enclosure into liposomes).